# Transcriptional Specificity Analysis of Testis and Epididymis Tissues in Donkey

**DOI:** 10.3390/genes13122339

**Published:** 2022-12-11

**Authors:** Mubin Yu, Xiaoyuan Zhang, Jiamao Yan, Jianhua Guo, Fali Zhang, Kexin Zhu, Shuqin Liu, Yujiang Sun, Wei Shen, Junjie Wang

**Affiliations:** 1College of Life Sciences, Qingdao Agricultural University, Qingdao 266109, China; 2Jiaozhou Agricultural and Rural Bureau, Jiaozhou 266300, China; 3College of Animal Science and Technology, Qingdao Agricultural University, Qingdao 266109, China; 4Key Laboratory of Animal Reproduction and Germplasm Enhancement in Universities of Shandong, Qingdao Agricultural University, Qingdao 266109, China

**Keywords:** testis, epididymis, transcriptome, WGCNA, donkey

## Abstract

Donkeys, with high economic value for meat, skin and milk production, are important livestock. However, the current insights into reproduction of donkeys are far from enough. To obtain a deeper understanding, the differential expression analysis and weighted gene co-expression network analysis (WGCNA) of transcriptomic data of testicular and epididymis tissues in donkeys were performed. In the result, there were 4313 differentially expressed genes (DEGs) in the two tissues, including 2047 enriched in testicular tissue and 2266 in epididymis tissue. WGCNA identified 1081 hub genes associated with testis development and 6110 genes with epididymal development. Next, the tissue-specific genes were identified with the above two methods, and the gene ontology (GO) analysis revealed that the epididymal-specific genes were associated with gonad development. On the other hand, the testis-specific genes were involved in the formation of sperm flagella, meiosis period, ciliary assembly, ciliary movement, etc. In addition, we found that eca-Mir-711 and eca-Mir-143 likely participated in regulating the development of epididymal tissue. Meanwhile, eca-Mir-429, eca-Mir-761, eca-Mir-200a, eca-Mir-191 and eca-Mir-200b potentially played an important role in regulating the development of testicular tissue. In short, these results will contribute to functional studies of the male reproductive trait in donkeys.

## 1. Introduction

Donkeys (*Equus asinus*), also called burro, ass or the domestic ass, belong to the equine family, and have been used as working animals for more than 5000 years. It has been reported by mitochondrial genomic analysis that the origin of the Chinese donkey has two branches, one originates from the Nubian wild ass, and the other is the Somali wild ass [1]. As one of the most common species used in agricultural production, the meat, skin and milk products in donkeys are of considerable commercial and economic value [2]. There are 40 million donkeys at least in the present world, mostly in underdeveloped countries, and they have always been indispensable livestock for people’s production and lives [3]. In view of the current situation, that the supply of donkey products is in short supply, it is extremely important to strengthen the conservation and breeding of donkeys, and to promote the research of their economic traits.

For the donkey species, the reproductive ability of male animals has a great impact on the development of the entire species. In-depth studies of its reproductive traits and good breeding work are necessary for the development of the entire industry. The production of sperm in mammals consists of three processes: differentiation and proliferation of spermatogonia, meiosis of spermatocytes and maturation of sperm [4]. Sperm production first begins from spermatogonia stem cells that produce spermatogonia in the seminiferous tubules, which then differentiate into primary spermatocytes. The secondary spermatogonia are formed after DNA replication and tetraploid meiosis, and finally spermatozoa are generated through chromatin condensation, acrosome formation, flagella formation and chromatin reduction. After maturation, spermatozoa are transferred to epididymis through the efferent duct for storage [5]. The whole process is controlled by the parallel and sequential action of numerous genes, and any abnormality in the expression of these genes can lead to spermatogenesis failure [6]. Studies have shown that the coiled-coil domain containing 136 (*CCDC136*), t-complex protein 11 (*TCP11*) and sperm autoantigenic protein 17 (*SP17*) genes are associated with spermatogenesis in male donkeys [7]. Among them, the testis-specific gene *CCDC136* plays an important role in sperm acrosome formation and fertilization. In *Ccdc136* knockout mice, sperm acrosomes are severely defective, resulting in reproductive disorders in males [8]. TCP11 interacts with outer dense fiber 1 (ODF1) and contributes to the formation of the unique morphology of the sperm tail [9]. In addition, the *SP17* gene functions in regulating sperm maturation, capacitation and interaction with the zona pellucida of oocytes [10]. Importantly, research on donkey spermatogenesis is fairly scant.

Noteworthily, the study of miRNAs involved in testicular development and spermatogenesis in male livestock presently attracted much attention [11,12]. Several non-coding microRNAs also play key roles in the post-transcriptional regulation of mRNA processing during spermatogenesis, as evidenced by recent studies in mammalian testis [13]. In male donkeys, the target gene of miR-449a, *COL6A2*, plays a role in Sertoli cells and Leydig cells, thereby affecting spermatogenesis and testosterone production [14]. miR-221 can activate the phosphatidylinositol 3-kinase (PI3K) signal transduction pathway through its target gene *ITGB1*, which is involved in embryonic testis spinal cord formation [15]. Moreover, the let-7 family may play a key role in the proliferation and differentiation of spermatogonia in horse [16]. Studies have shown that the inhibition of let-7 can block the proliferation and differentiation of spermatogonia [16]. In equine epididymal epithelial cells, it has been shown that miRNA indirectly affects sperm motility through target genes [17]. However, the regulatory mechanism of miRNA in spermatogenesis still needs further investigation.

Owing to the development of high-throughput sequencing technology, RNA sequencing (RNA-seq) has been applied widely in biological research by constructing gene-scale transcriptional maps. It provides a new research strategy for analyzing gene functions and miRNA regulatory mechanisms. In this study, the combined method of comparative transcriptomics and weighted gene co-expression network analysis (WGCNA) was used to pick out the tissue-specific genes and miRNAs which may play a regulatory effect during spermatogenesis, and provide the important clues of functional genes involving in male spermatogenesis of the donkey.

## 2. Materials and Methods

### 2.1. Data Collection

To identify specific transcriptomic characteristics related to donkey reproduction in males, we searched for relevant data in the National Center for Biotechnology Information (NCBI) database with testicular tissue and epididymal tissue as qualifiers. Three sets of transcriptome data, including donkey testis and epididymis tissues, as well as horse testis tissue, were downloaded and a total of 65.6 Gb of raw data was obtained. Accession numbers were listed as follows: donkey testis, donkey epididymis [PRJNA512590, PRJNA431818] [7,18], horse testis [PRJNA509608, PRJNA395221] (Appendix A) [4,19].

### 2.2. Quality Control of Data

To ensure the consistency of the data analysis, the raw data were downloaded and proceeded following the same standard. Quality control was checked according to the report formed by FastQC (v0.11.8) [20]. FASTP (v0.23.1) software was performed to remove low-quality data, including sequencing adapters and unqualified data [21].

### 2.3. Genome Mapping and Calculation of Gene Expression Level

Subsequently, the high-quality data were aligned to the reference genome with STAR (v2.7.0f) [22] software, including the donkey reference genome (GCA_003033725.1) and the horse reference genome (GCA_002863925.1), and then we used FeatureCounts (v1.6.3) software for conducting reads counts [23]. As for miRNA data, we extracted the miRNA information from the reference genome, then the FeatureCounts software was employed to calculate miRNA count information.

### 2.4. Principal Component Analysis (PCA) and Hierarchical Clustering Analysis

To gain the clustering information between samples, we normalized the data matrix using the *rlog* function of DESeq2 (R package, v1.36.0), then PCA was performed by using the *plotPCA* function [24]. In addition, we used a custom R script to perform a hierarchical clustering analysis and visualized it by the pheatmap (R package, v1.0.12).

### 2.5. Identification of Differentially Expressed Genes (DEGs) and miRNAs

The DESeq2 was applied to screen DEGs between different tissues, including donkey testis, donkey epididymis and horse testis. The DEGs were determined by the criteria of “|Log2(fold change)| > 2 and *p*-adjust < 0.01” [24]. To obtain differentially expressed miRNAs (DE-miRNAs), we used DESeq2 to compare the miRNAs counts with samples of two groups, “|Log2(fold change)| > 0.5 and *p*-value < 0.05” were considered as the criteria of DE-miRNAs, and the differential miRNA results were displayed using volcano plots [24].

### 2.6. Weighted Gene Co-Expression Network Analysis (WGCNA)

The WGCNA was performed by using the WGCNA package (v1.71) [25]. In this part, the data of three tissues, including donkey testis, donkey epididymis and horse testis, were employed with a total of 16 samples. First, the FPKM was chosen to normalize gene expression levels. Next, we built a scale-free network with the *BlockWiseModules* function. Specifically, the parameters of ‘*power = 5, MinModuleSize = 30 and MergeCutheight = 0.3′* were applied to determine gene modules. Further, the correlation between tissue specificity and gene modules was explored to find the hub genes, which are considered key nodes in modules and are highly interconnected. Here, we used the module membership (MM) and gene significance (GS) methods to evaluate hub genes in modules, with thresholds of MM > 0.8 and GS > 0.8 [26].

### 2.7. Construction of Tissue-Specific Regulatory Network

Based on the result of tissue-enriched genes (DEGs) and hub genes, the R package VennDiagram (v 1.7.3) was used to intersect these genes, and the genes in the overlapping part of the Venn diagrams were termed as tissue-specific expressed genes. Then the tissue-specific expressed genes were subjected to functional analysis, the terms related to reproduction were screened out, and finally the tissue-specific regulatory network graphs containing terms and their enriched genes were built by Cytoscape (v 3.9.1) [27].

### 2.8. Annotation of DEGs and DE-miRNAs, Prediction of DE-miRNAs Target Genes

To gain insight into the regulatory role of DEGs and DE-miRNAs in male donkey reproductive organs, we used TargetScan (http://www.targetscan.org/vert_80/, accessed on 21 September 2022) and miRDB (http://www.mirdb.org/, accessed on 21 September 2022) software for the prediction of differential miRNA target genes. Due to the lack of donkey and horse target gene prediction databases, we used g:Profiler to convert donkey and horse gene IDs to homologous IDs of mice [28].

### 2.9. Functional Annotation Analysis of Genes

We performed Gene Ontology (GO) and Kyoto Encyclopedia of Genes and Genomes (KEGG) analyses for DEGs, hub genes and DE-miRNAs target genes using KOBAS [29]. A *p*-value < 0.05 was considered to be statistically significant.

## 3. Results

### 3.1. An Overview of Transcriptome Data with Reproductive Tissues in Donkey and Horse

To investigate the tissue-specific characteristics of the transcriptome in donkeys, we collected RNA-seq data from donkey testis and epididymis tissues, as well as horse testis tissue. Subsequently, followed by the data processing, the comparative analysis and WGCNA were applied to reveal the tissue-specific genes and miRNAs (Figure 1A). Firstly, the hierarchical clustering analysis showed that the sample repeatability was good, and the transcriptomic difference between testis and epididymis tissues in the two species was relatively more distinct than that of testis vs. epididymis tissues in the same species (Figure 1B). Meanwhile, the PCA suggested a distinct distance of data between the testis and epididymis in donkeys (Figure 1C), as it did for testis tissues between donkey and horse (Figure 1D). Moreover, compared with testis in donkeys, it indicated that the dispersion of epididymis data in donkeys is much larger than that of testis samples in the horses (Figure 1C,D).

### 3.2. Differential Expression Analysis of Transcriptomic Data

Here, the tissue-enriched genes were screened through differential expression analysis. In the result of comparative grouping between testis and epididymis tissues in donkeys, a total of 4313 DEGs were found, among which, 2047 DEGs (up-DEGs) were enriched in donkey testis tissue, and 2266 DEGs (down-DEGs) were enriched in donkey epididymis tissue (Figure 2A). In the result of testis tissue between donkey and horse, there were 1811 DEGs in total, of which 883 DEGs (up-DEGs) were enriched in the horses, and 928 DEGs (down-DEGs) were enriched in the donkey (Figure 2B). Next, the top 200 DEGs in each group were selected for GO analysis based on the value of |Log2(fold change)|. In terms of DEGs between testis and epididymis tissues in donkeys, we found 310 significant GO terms enriched in donkey testis tissue, and the top five enriched terms were mainly related to the processes of “integral component of membrane”, “membrane”, “sequence-specific double-stranded DNA binding”, “plasma membrane” and “G protein-coupled receptor signaling pathway” (Figure 2C, Appendix A). In donkey epididymis, 539 significant GO terms were enriched, and the top five enriched terms were “extracellular region”, “extracellular space”, “sequence-specific DNA binding”, “sequence-specific double-stranded DNA binding” and “embryonic skeletal system morphogenesis” (Figure 2D, Appendix A). As for the DEGs of the testis between donkey and horse, 324 significant GO terms were enriched in the testis of the horse, and the top five enriched biological processes included “plasma membrane”, “integral component of membrane”, “membrane”, “extracellular region” and “negative regulation of translation” (Figure 2E, Appendix A). In donkey testis tissue, 391 significant GO terms were enriched, and the top five enriched terms were involved in the biological processes of “nucleus”, “protein binding”, “membrane”, “cytoplasm” and “integral component of membrane” (Figure 2F, Appendix A). Meanwhile, the genes enriched in the terms were also displayed.

### 3.3. WGCNA Identified the Tissue-Specific Module Genes in Donkey

To find tissue-specific genes associated with testis and epididymis in donkeys, we performed WGCNA using the FPKM matrices from all samples. With a soft threshold setting for a scale-free network of gene expression matrix, all the genes were divided into 14 modules, of which the turquoise had the most genes, reaching 9097, and the salmon had the least, with only 31 genes (Figure 3A). Meanwhile, it showed that some modules were correlated with tissue specificity, in particular, the turquoise module was related to donkey epididymis tissue, the brown module to donkey testis and the blue module to horse testis. Further, the hub genes in the modules were screened by MM > 0.8 and GS > 0.8. In the turquoise module, 6110 hub genes were identified. There were 1081 hub genes in the brown module; 2201 hub genes were included in the blue module (Figure 3A,B). With regard to the hub genes of the turquoise module, they were mainly related to the biological processes of “spermatogenesis”, “flagellated sperm motility”, “cilium assembly”, “meiotic cell cycle”, “sperm flagellum”, etc. (Figure 3C). In the brown module, the top five terms included “cilium assembly”, “spermatogenesis”, “sperm axoneme assembly”, “DNA methylation involved in gamete generation”, “sperm head”, etc. (Figure 3D). Finally, the hub genes in the blue module were most enriched in the terms of “spermatogenesis”, “cilium assembly”, “sperm flagellum”, “flagellated sperm motility”, “motile cilium”, etc. (Figure 3E).

To identify the tissue-specific genes, we combined the result of hub genes and enriched genes in tissues, and the overlapping genes were viewed as tissue-specific genes. We found 1459 tissue-specific genes in donkey epididymal tissue (Figure 4A), 363 tissue-specific genes in donkey testis tissue (Figure 4B) and 381 tissue-specific genes in horse testis tissue (Appendix A). The tissue-specific genes were subjected to functional enrichment analysis. After that, we found 11 terms that were related to the function of epididymis tissue in the donkey. They were “male gonad development”, “male genitalia development”, “reproductive process”, “reproductive structure development”, “single fertilization”, “male genitalia morphogenesis”, “sperm ejaculation”, “sperm-egg recognition”, “fusion of sperm to egg plasma membrane involved in single fertilization”, “regulation of meiotic nuclear division” and “positive regulation of male gonad development”, and the genes enriched in the terms are listed (Figure 4C). In donkey testis tissue, 10 terms and a total of 24 tissue-specific genes were displayed including “cilium”, “cilium movement”, “motile cilium”, “sperm axoneme assembly”, “male meiosis I”, “sperm flagellum”, “cilium assembly”, “motile cilium assembly”, “homologous chromosome pairing at meiosis” and “flagellated sperm motility” (Figure 4D). In horse testis tissue, five terms and a total of 17 tissue-specific genes enriched in the processes of “sperm axoneme assembly”, “male meiotic nuclear division”, “single fertilization”, “spermatogenesis” and “male gonad development” (Appendix A). In addition, we constructed a mixed pool of reproductive-related genes specifically expressed in the three tissues, and these genes were highly expressed in the corresponding tissues. (Figure 4E and Appendix A).

### 3.4. The Identification of miRNAs Related to Tissue Specificity in Donkey

Further, the tissue-enriched miRNAs (TE-miRNAs) in donkeys were investigated. Through differential expression analysis, we obtained miRNAs enriched in donkey testis and epididymis tissues. It showed that 30 miRNAs (up-miRNAs) were significantly enriched in the testis tissue of donkeys, and 25 miRNAs (down-miRNAs) in epididymis tissue (Figure 5A). In the comparison of testis tissue between donkey and horse, we found a total of 39 TE-miRNAs, including 17 miRNAs (up-miRNAs) in horse and 22 miRNAs (down-miRNAs) in donkey (Figure 5B). In order to explore the regulatory role of miRNAs, we carried out target gene prediction of miRNAs. Due to the limitation of the database, only two miRNAs can be annotated with predicted target genes in donkey testis tissue, and six miRNAs in donkey epididymis tissue (Table 1). In terms of testis tissues between horse and donkey, 3 miRNAs were annotated in horse testis tissue and 10 miRNAs in donkey testis tissue (Table 2). Based on the TE-miRNAs in the tissues, we predicted 1516 target genes in the testis tissue of the donkey, 189 target genes in the epididymis of the donkey and 773 target genes in the testis of the horse (Appendix A). Then, the target genes of miRNAs in the testis of the donkey were obtained, with 2009 terms significantly (*p*-value < 0.05) enriched, and the top five enriched terms were “protein binding”, “cytoplasm”, “nucleus”, “membrane” and “cytosol” (Figure 5C, Appendix A). In donkey epididymis tissue, 684 terms were obtained (*p*-value < 0.05), and the top five enriched terms were mainly related to “protein binding”, “cytoplasm”, “nucleus”, “membrane” and “plasma membrane” (Figure 5D, Appendix A). In the comparison of testis tissue between horse and donkey, we obtained 1193 terms in horse testis tissue (*p*-value < 0.05), and the top five enriched terms were mainly related to “protein binding”, “membrane”, “nucleus”, “cytoplasm” and “plasma membrane” (Figure 5E, Appendix A). Further, we obtained 2490 pathways in donkey testis tissue (*p*-value < 0.05), they included “protein binding”, “cytoplasm”, “nucleus”, “membrane” and “nucleoplasm” (Figure 5F, Appendix A).

Importantly, we focused on the overlapping parts of miRNA target genes and tissue-specific genes for follow-up analysis, and the overlapping parts called key genes. The results showed that in donkey testis tissue, there were 25 key genes overlapped by testis-specific genes and targeted genes of miRNAs (Figure 6A), and they were regulated by miR-429-3p, miR-761, miR-200a-3p, miR-191-5p and miR-200b-3p (Appendix A). In addition, these genes were enriched in the pathways of “cell cycle”, “cellular senescence”, “microRNAs in cancer”, “bladder cancer”, “cocaine addiction”, “non-small cell lung cancer”, “melanoma”, “inositol phosphate metabolism”, “glioma”, “pancreatic cancer” and “chronic myeloid leukemia”, and the potential relationship of miRNA regulating these pathways via target genes (including *CDC25A*, *CDK5R1*, *E2F3* and *SYNJ1*) was displayed in the Sankey diagram (Figure 6B). In donkey epididymis tissue, miR-143-3p, miR-711, miR-200b-3p, miR-761 and miR-429-3p regulated 18 genes (Figure 6C, Appendix A), and the genes were enriched in the pathway of “TGF-β signaling pathway”, “mucin type o-glycan biosynthesis” and “melanoma”, which may be regulated by miR-143-3p and miR-711 through *BMP5*, *FGF7*, *GCNT1* and *THSD4* (Figure 6D). In horse testis tissue, 10 key genes were uncovered by differential expression analysis and WGCNA (Appendix A), which were regulated by miR-214-5P, miR-375-3p and miR-761 (Appendix A). miR-761 might be involved in multiple pathways through *GABRA4*, *PLA2G12A* and *UGGT2*, such as “α-linolenic acid metabolism”, “glycerophospholipid metabolism”, “pancreatic secretion”, “retrograde endocannabinoid signaling” and “protein processing in endoplasmic reticulum” (Appendix A).

## 4. Discussion

Reproductive traits have always been a hotspot in the field of animal husbandry. As one of the most important livestock in the world, research focusing on the domestic donkeys still scant. In the present study, based on the transcriptome data, with the help of differential expression analysis and WGCNA, the tissue-specific expression genes and miRNA were deeply analyzed in the testis and epididymis of the donkey. At the same time, it revealed the potential regulatory relationship of miRNAs on some biological processes in the testis and epididymis. Our research screened numerous genes associated with reproductive tissue in male donkeys, which may provide important clues about candidate genes involved in the regulatory mechanism for reproductive traits of the donkey.

In our study, among the overlapping genes between testis tissue-enriched genes and hub genes in donkeys, three highly expressed genes in the spermatogenesis process were detected, including *REC8*, *DNMT3L* and *FANCD2*. During meiotic prophase, the rec8-dependent axis-loop structure is essential for linear element assembly, which plays an important role in the meiotic recombination process of homologous chromosomes [30]. The *DNMT3L* histone H3-binding domain (ADD) is required for spermatogenesis, and plays an important role in the DNA methylation of some genes during spermatogenesis, essential for spermatozoa maturation in males [31,32]. At the same time, we also found that *DNAH1* and *RSPH6A* genes related to sperm motility in the testis. The *DNAH1* gene is involved in the formation of the internal power arm of sperm, and its deletion can lead to decreased sperm motility and male infertility [33]. *Rsph6a* is essential for sperm flagella assembly and male fertility in mice. Studies have shown that after knocking out *Rsph6a* in mice, the axons of sperm flagella can be elongated, but mitochondria and fibrous sheaths are abnormally formed, and sperm flagella are severely defective [34]. During the transfer of sperm from the testis to the epididymis, we found two genes involved in this process, including *CCNO* and *CEP164*. The atypical cyclin (CCNO) is required for the generation of functional polyciliated cells in the efferent duct. In mice with mutations in the *Ccno* gene, the function of the efferent ducts is impaired, preventing sperm from entering the epididymis, and causing reproductive impairment [35]. The deletion of the distal appendage protein (CEP164) leads to the loss of multifunctional cilia in the efferent ducts and triggers sperm aggregation in efferent ducts, causing reproductive dysfunction [36].

In epididymal tissue-specific genes, we focused on “regulation of meiotic nuclear division”, “sperm-egg recognition” and “single fertilization”. In the process of “regulation of meiotic nuclear division”, we found *CDH1* and *PDE3A* genes. E-cadherin (epithelial cadherin), encoded by the *Cdh1* gene, which plays a role in gonad development, enables adhesion between germ cells and somatic cells to ensure proper signaling between somatic and germ cells, which is essential for germ cell survival [37]. The PDE3A is a member of the phosphodiesterase family, located in the post-acrosomal segment of the sperm head, and regulates the motility, capacitation and acrosome reaction of mammalian sperm through a signal transduction system [38,39]. In terms of sperm and egg recognition and combination, we found the *CD9* and *RNASE10* genes. Interaction of sperm surface proteins with CD9 or CD9-related oocyte proteins has the potential to trigger PTK2B activation at sperm binding sites, and PTK2B kinase activity plays an important role in actin remodeling at sperm binding sites and promotes sperm incorporation [40]. Epididymal protein Rnase10 is involved in the processing and stabilization of Adam3 protein (an adhesion protein of sperm to egg). In sperm serum, *Rnase10* deficiency leads to the inability of sperm to recognize the egg [41].

In the follow-up study to explore the regulatory relationship between DE-miRNAs and tissue-specific expression of DEGs, we found two joint node genes in testis tissue, namely *E2F3* and *CDC25A* (Cell division cycle 25A). In the study, the *E2F3* gene is regulated by eca-miR-200a. It has been reported that E2F3 exists in the promoter region and plays a regulatory role in the process of the cell cycle from G1 to S phase. At the same time, it inhibits the production of fat by inhibiting the binding of CEBPA to the promoter of its target gene [42]. In addition, *CDC25A* is a key factor in regulating the cell cycle and is regulated by eca-miR-200a. Mutual feedback regulation between CDC25A and DYRK2 has important implications for genotoxic stress pathways, apoptosis and cell cycle regulation [43]. In epididymal tissue, four genes were demonstrated with miRNA roles, including *BMP5*, *GCNT1*, *FGF7* and *THSD4*. Among them, *BMP5*, regulated by eca-miR-143, is a member of the TGF-β superfamily, and plays an important role in male reproductive function [44]. *GCNT1*, regulated by eca-miR-143, is induced by androgens in prostate cancer (PC) cells at the protein level, and the production of GCNT1 is important for the synthesis of cancer-associated O-glycans and contributes to PC cell viability [45]. *FGF7* is regulated by eca-miR-143, and belongs to the FGF family; it has a wide range of activities in embryonic development and physiological functions in adults. [46]. *THSD4* is regulated by eca-miR-711, and it may be a potential regulator of prostate cancer. It is involved in tumorigenesis and the development of various cancers, including breast, glioblastoma and esophageal cancer [47]. Actually, the role of miRNA in male reproductive development is worthy of deep exploration.

In the present study, the transcriptome data of donkey testis and epididymis tissues were employed to screen out the tissue-specific genes and TE-miRNAs, which provide important information for the potential regulatory mechanism of male reproductive traits in the donkey. At the same time, the horse transcriptome data were also collected for joint analysis and it aimed to identify the conserved genes in testicular tissue between donkey and horse species, which contributes to subsequent studies on functional genes. However, it is noted that certain limitations cannot be ignored in the results of the study. One is the influence of age and feeding conditions in conservation analysis. For large livestock, it is difficult to collect samples at the same age and under the same conditions of the breeding environment. Additionally, the transcriptome data used in the study are derived from different research projects, thus, it may weaken the reliability of results. The second issue is that the result of the study is mainly the analyses of transcriptome data and it is lack of experimental validation. Thus, we believe that it is necessary to further verify the conservation analysis between donkey and horse species, as well as the functional gene studies of the male reproductive traits in the donkey.

## 5. Conclusions

In this study, a gene-scale transcriptional map of male donkeys was constructed by using RNA sequencing data. Through the combination of comparative transcriptome analysis and WGCNA, the genes and miRNAs related to testicular and epididymis tissues were identified, which may be involved in the regulatory events during spermatogenesis, including sperm maturation and storage in the donkey. It provides data supporting functional gene studies of the male reproductive traits in male donkeys.

## Figures and Tables

**Figure 1 genes-13-02339-f001:**
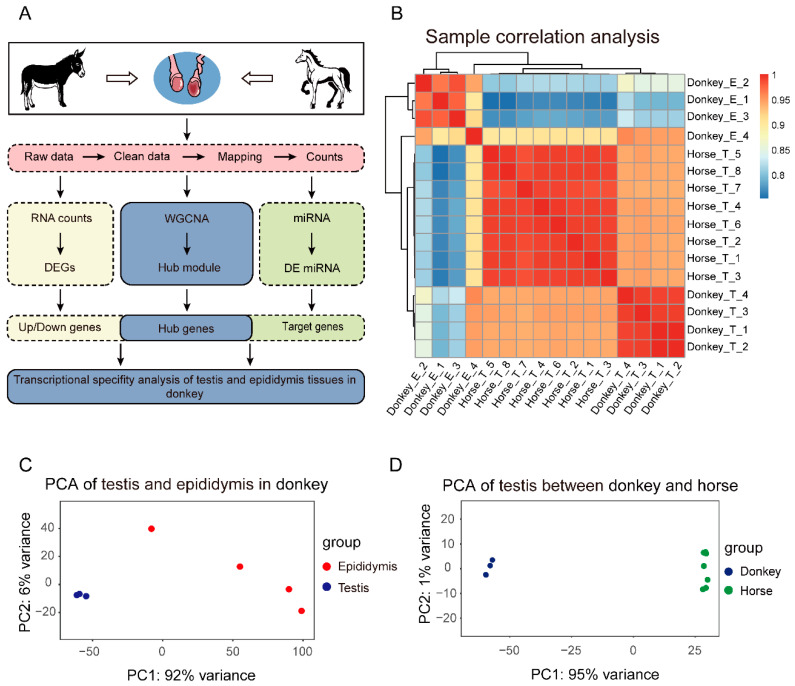
The gene expression landscape based on the transcriptome data of different tissues. (**A**) The research flow chart of collected transcriptome data in this study. (**B**) Hierarchical clustering of gene expression matrices for all samples including testis and epididymal tissues of the donkey, and horse testis tissue. (**C**) Principal component analysis (PCA) of donkey testis and epididymis tissues. (**D**) PCA of testis tissues between donkey and horse.

**Figure 2 genes-13-02339-f002:**
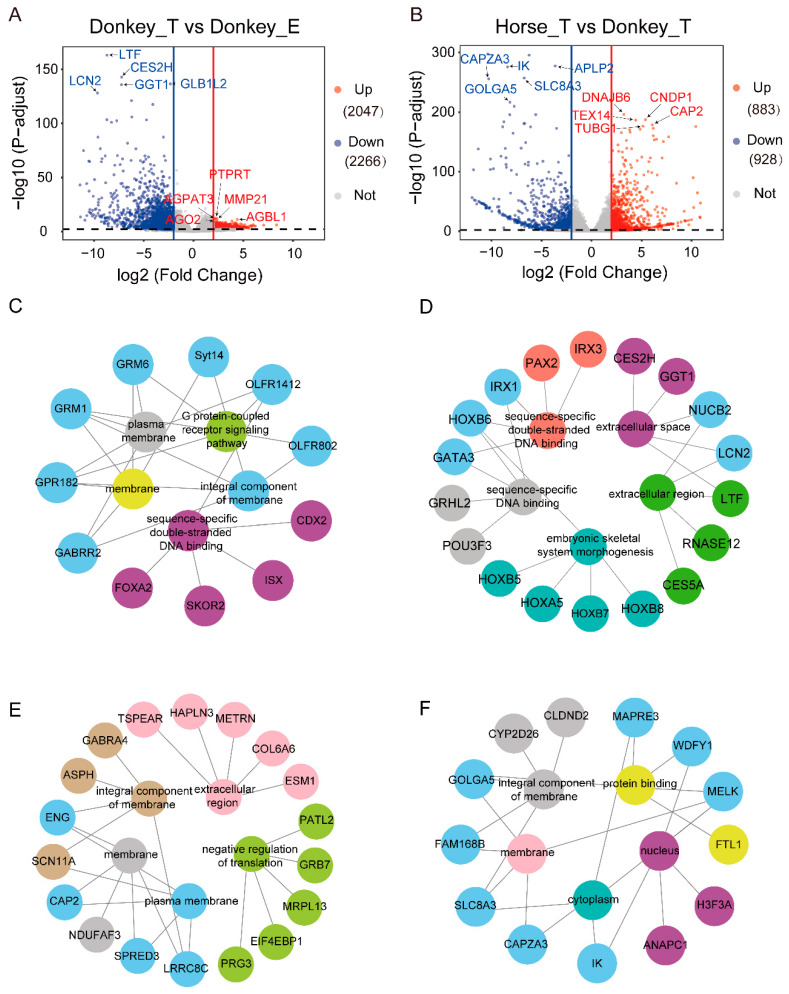
Identification of the tissue–enriched genes in donkeys and horses. (**A**) Volcano plot of DEGs between testis and epididymis tissues in the donkey. Red indicates up–regulated DEGs (up–DEGs); blue indicates down–regulated (down–DEGs). The marked genes are the top five selected by *p*-adjusted value. (**B**) Volcano plot of the differentially expressed genes in testis tissue between donkey and horse. Red indicates up–regulated DEGs (up–DEGs); blue indicates down–regulated DEGs (down–DEGs). The marked genes are the top five selected by *p*-adjusted value. (**C**) The inner circle shows the top five GO (biological process) terms for testis tissue–enriched genes in donkeys, and the outer circle shows the genes enriched in the terms. (**D**) The top five GO terms of tissue–enriched genes in donkey epididymis. (**E**) The top five GO terms of tissue–enriched genes in horse testis (**F**) The top five GO terms of tissue–enriched genes in donkey testis. The GO terms refer to the biological processes.

**Figure 3 genes-13-02339-f003:**
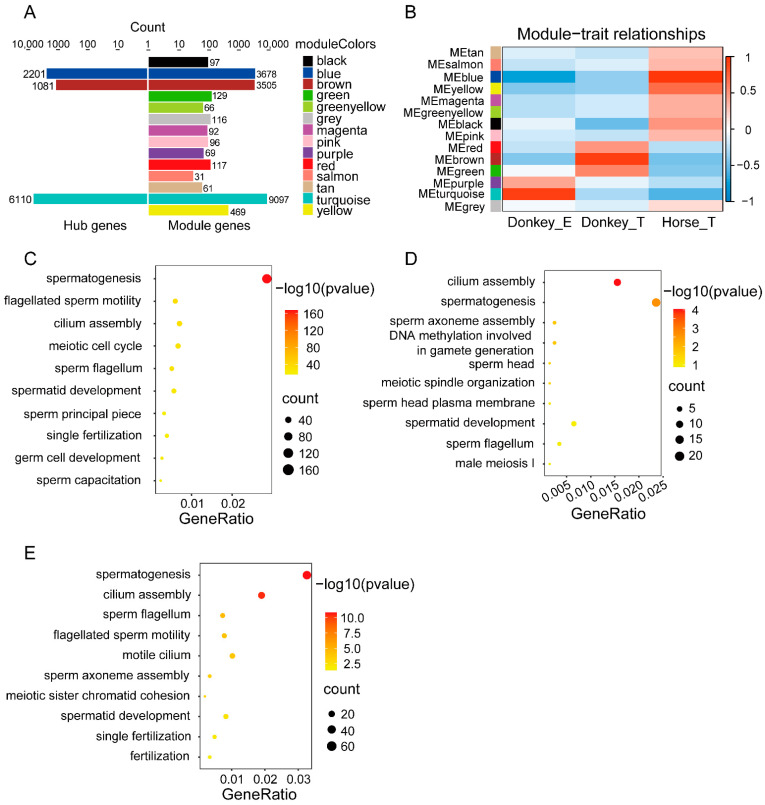
Weighted gene co–expression network analysis (WGCNA) of different tissues. (**A**) The histogram shows the number of genes in the modules identified by WGCNA; the right panel shows the number of genes in the modules and the left panel shows the number of hub genes. (**B**) Heatmap of module–tissue correlations. Red indicates a strong correlation and green indicates a weak correlation. (**C**) Bubble plots represent biological processes enriched by hub genes in donkey epididymal tissue. (**D**) Bubble diagram of biological process enriched by hub genes in donkey testis tissue. (**E**) Bubble diagram showing biological process enriched by hub genes in horse testis tissue.

**Figure 4 genes-13-02339-f004:**
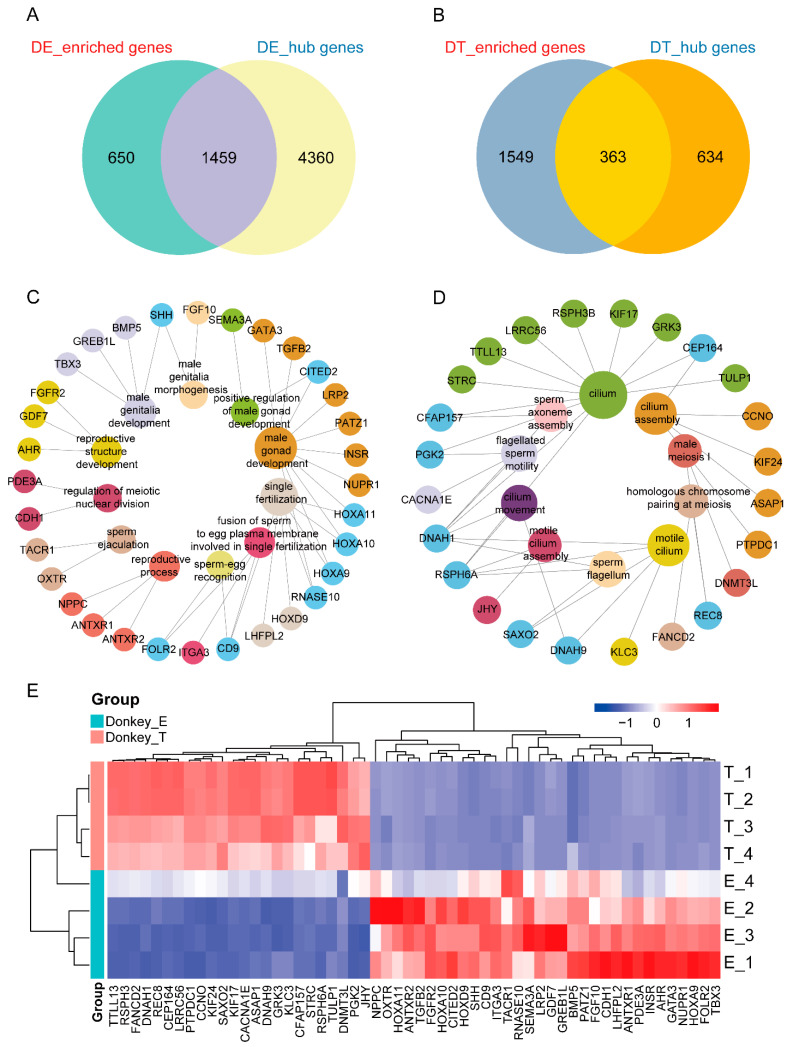
Analysis of tissue–specific expressed genes in the donkey. (**A**) Venn diagram of tissue–specific genes in donkey epididymal tissue. The overlap genes between tissue–enriched genes and hub genes in donkey epididymal tissue were called tissue–specific genes. (**B**) Venn diagram of tissue–specific genes in donkey testis tissue. (**C**) The inner circle shows GO (biological process) terms related to reproductive terms enriched by tissue–specific genes in donkey epididymal tissue, and the outer circle shows genes enriched in the terms. (**D**) The inner circle shows the GO (biological process) terms related to reproductive terms enriched by tissue–specific genes in donkey testis tissue, and the outer circle shows genes enriched in the terms. (**E**) The heat map shows the expression of tissue–specific genes in the testis and epididymal tissues of the donkey; red represents high expression and blue represents low expression.

**Figure 5 genes-13-02339-f005:**
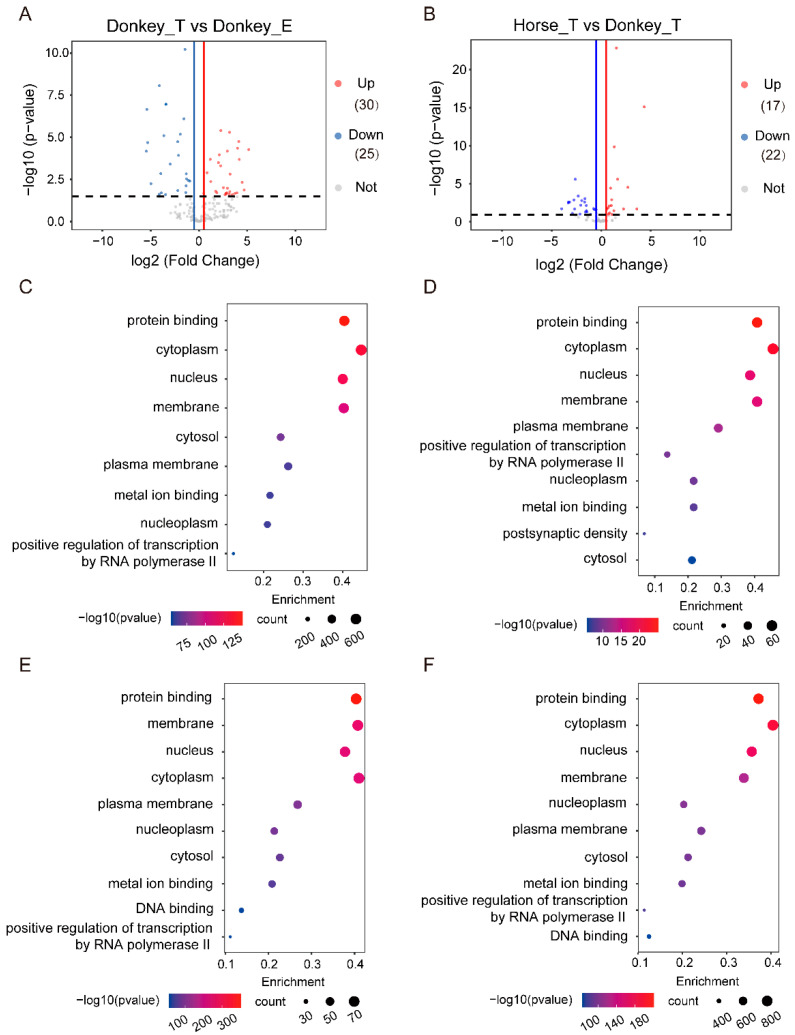
Functional analysis of tissue–specific miRNAs in testis and epididymis of the donkey. (**A**) Volcano plot of differentially expressed miRNAs in testis and epididymis tissues of the donkey. (**B**) Volcano plot of differentially expressed miRNAs in testis tissues of donkey and horse. (**C**) The top five GO (biological process) terms of targeted genes of miRNAs enriched in donkey testis tissue. (**D**) The top five GO terms (biological processes) of targeted genes of miRNAs enriched in donkey epididymal tissue. (**E**) The top five GO (biological process) terms of targeted genes of miRNAs enriched in horse testis tissue. (**F**) The top five GO (biological process) terms of targeted genes of miRNAs enriched in donkey testis tissue.

**Figure 6 genes-13-02339-f006:**
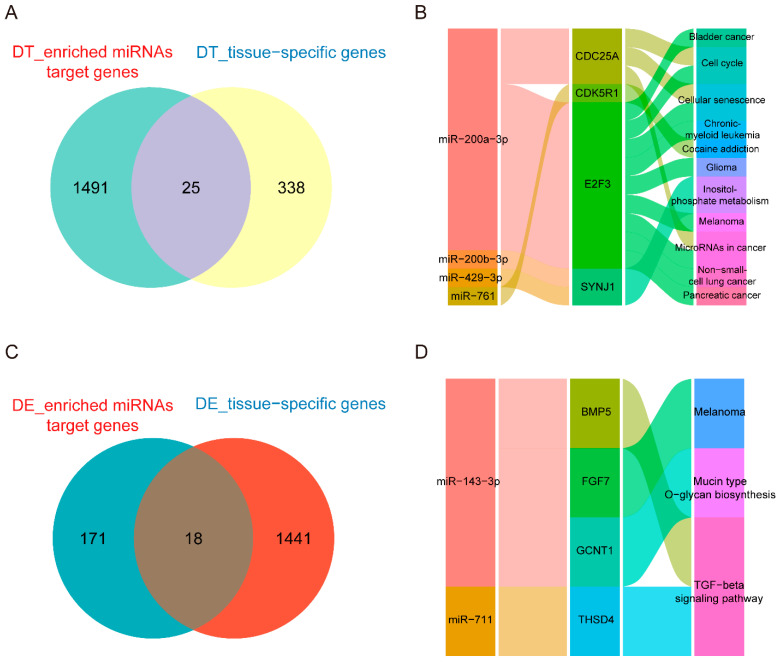
The regulatory analysis of miRNAs on pathways through tissue key genes in the donkey. (**A**) Venn diagram of key genes in donkey testis. The overlapping gene between target genes of tissue-enriched miRNAs and tissue–specific expressed genes in donkey testis were called donkey testis key genes. (**B**) Sankey diagram shows the KEGG pathway regulated by miRNAs through key genes in donkey testis. (**C**) Venn diagram of key genes in donkey epididymis. The overlapping genes between target genes of tissue-enriched miRNAs and tissue–specific expressed genes in donkey epididymis tissue were called donkey epididymis key genes. (**D**) Sankey diagram shows the KEGG pathway regulated by miRNA through key genes in donkey epididymis.

**Table 1 genes-13-02339-t001:** Annotation of enriched-miRNAs in testis and epididymis tissues of donkey.

Types	Enriched_miRNAs	
**Testis**	**eca-mir-711**	**eca-mir-143**
	eca-mir-9182	eca-mir-9092
	eca-mir-8999	eca-mir-8917
	eca-mir-8914	eca-mir-664
	eca-mir-1902	
**Epididymis**	**eca-mir-761**	**eca-mir-429**
	**eca-mir-375**	**eca-mir-200b**
	**eca-mir-3548**	**eca-mir-191b**
	eca-mir-9141	eca-mir-9120
	eca-mir-9087	eca-mir-9054
	eca-mir-9032	eca-mir-8990
	eca-mir-8966	eca-mir-632
	eca-mir-1905c	eca-mir-1842
	eca-mir-1307	

**Table 2 genes-13-02339-t002:** Annotation of enriched-miRNAs in horse testis tissue and donkey testis tissue.

Types	Enriched_miRNAs		
**Horse**	**eca-mir-375**	**eca-mir-761**	**eca-mir-214**
	eca-mir-1307	eca-mir-8965	eca-mir-1842
	eca-mir-7035	eca-mir-9063	eca-mir-9146
	eca-mir-1282	eca-mir-9032	eca-mir-9054
	eca-mir-9010	eca-mir-9056	eca-mir-1892
	eca-mir-9035	eca-mir-9120	
**Donkey**	**eca-mir-149**	**eca-mir-27b**	**eca-mir-24**
	**eca-mir-155**	**eca-mir-200b**	**eca-mir-145a**
	**eca-mir-429**	**eca-mir-143**	**eca-mir-20a**
	**eca-mir-200a**	eca-mir-324	eca-mir-1905c
	eca-mir-1291a	eca-mir-1248	eca-mir-9140
	eca-mir-9141	eca-mir-9102	eca-mir-9055
	eca-mir-9086	eca-mir-9089-1	

## Data Availability

All data generated or analyzed during this study are included in this article.

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
