# Peer review of "Transcriptional Specificity Analysis of Testis and Epididymis Tissues in Donkey"

_genes, 2022, doi:10.3390/genes13122339_

Round 1

Reviewer 1 Report

The study analyzed the transcriptional profile of testis and epididymis tissue in donkey. The authors of the study used the transcriptional data deposited in the gene project repository. The work resulted in identifying the most expressed genes, DEGs, hub genes, and the tissue-enriched miRNAs (DE-miRNAs) in each tissue; there were also identified GO terms enriched as well as  KEGG processes in tissues studied. This compelling study used Weighted gene co-expression network analysis (WGCNA) to identify hub genes.

The research is interesting but requires modification in some points

1. In my opinion the aim of the study is not clearly defined (lines 80-84). Firstly neither the title, Abstract nor Introduction lines 80-84 mentioned about horse tissues. It is not explained why the authors decided to introduce horse tissue data to the study.

2. The number of donkey samples studied (n=4) is not enough. Especially since the transcriptomic data analyzed in the manuscript has been obtained from two different research projects. This means that samples were probably collected from animals of different ages and feeding differently. I have read the manuscript of Tian et al. 2020 (doi:10.1016/j.livsci.2019.103885) and Wang 2020 (doi:10.1038/s41467-020-462 19813-7). In the first manuscript was studied RNA-seq on samples from three donkeys. The authors have not provided details about the age or feed of the animals. The second reference manuscript regards with resequencing of the donkey genome. The research was conducted on genomic DNA. This means that the authors of the reviewed manuscript the data obtained from gene expression and genome DNA projects and the results presented in GO, KEGG, or DEGs outputs are difficult to word any certain conclusion. This needs more detailed comments from the authors.

3. In Results are not listed the high expressed genes, but the GO and KEGG processes are listed in detail. I think that the genes with the greatest impact on each biological process should be listed in this section. The genes are listed in the Supplementary file, but there is no any information if they are listed hierarchically and each tissue they are regarding.

4. The description of Figures sometimes needs improvement: Figure 2 regards DEGs in donkey and Horse.

5. I recommend English revision of the manuscript (line  43: “very necessary”, line228: :It was found that we found”

Reviewer 2 Report

Good paper. Publish.

Please I like know:

What software did you use to make figure 4 (A, B, C, D). How can you organize the dataset (can you show me the data structure?)

Thanks, 

Round 2

Reviewer 1 Report

All of my previous comments have been taken into account. Although the authors provided their comments in response to the reviewer, I would like to see these comments in the manuscript text too. This concerns in particular the precise aim of the study and explanation of using horse tissues. I have not found Supplementary table 3 in the text.

After doing these minor corrections I recommend the manuscript for publication.
